# Research on Electrostatic Field-Induced Discharge Energy in Conventional Micro EDM

**DOI:** 10.3390/ma16113963

**Published:** 2023-05-25

**Authors:** Yaou Zhang, Qiang Gao, Xiangjun Yang, Qian Zheng, Wansheng Zhao

**Affiliations:** 1State Key Laboratory of Mechanical System and Vibration, Shanghai 200240, China121020920638@sjtu.edu.cn (X.Y.); zhengqian@sjtu.edu.cn (Q.Z.);; 2School of Mechanical Engineering, Shanghai Jiao Tong University, Shanghai 200240, China

**Keywords:** field-induced discharge, energy generation method, electrolyte jet electrode, discharge in gas, micro EDM

## Abstract

The electrostatic field-induced electrolyte jet (E-Jet) electric discharge machining (EDM) is a newly developed micro machining method. However, the strong coupling of the electrolyte jet liquid electrode and the electrostatic induced energy prohibited it from utilization in conventional EDM process. In this study, the method with two discharge devices connecting in serials is proposed to decouple pulse energy from the E-Jet EDM process. By automatic breakdown between the E-Jet tip and the auxiliary electrode in the first device, the pulsed discharge between the solid electrode and the solid workpiece in the second device can be generated. With this method, the induced charges on the E-Jet tip can indirectly regulate the discharge between the solid electrodes, giving a new pulse discharge energy generation method for traditional micro EDM. The pulsed variation of current and voltage generated during the discharge process in conventional EDM process verified the feasibility of this decoupling approach. The influence of the distance between the jet tip and the electrode, as well as the gap between the solid electrode and the work-piece, on the pulsed energy, demonstrates that the gap servo control method is applicable. Experiments with single points and grooves indicate the machining ability of this new energy generation method.

## 1. Introduction

Micro EDM (electrical discharge machining) is a specialized machining method that processes conductive materials by using the thermal energy produced by pulse discharges between two electrodes, which melts and vaporizes the material from the workpiece as well as the tool electrode. This method yields numerous benefits. For example, EDM does not make direct contact between the electrode and the workpiece eliminating mechanical stresses, chatter, and vibration problems during machining [1], and the hardness of the tool electrode material may be lower than that of the workpiece material. It can process materials that are difficult to deal with by traditional methods [2], and is particularly suited for small and complex features in aviation [3], diesel engines [4], nuclear power plants [5], and medical [6] and mold [7] manufacturing industries.

The discharge energy is significant because it is intimately related to material removal rate, machining accuracy, and surface integrity. In micro EDM, the small pulse energy is one of the prominent research topics. The major discharge energy generation methods are the relaxation pulse generator [8] and the transistor-pulse generator [9]. Singh et al. [8] verified that the relaxation pulse generator can generate low discharge energy by charging and discharging the capacitor adopted in the main power loop. Ichikawa and Natsu [10] further confirmed that the existing stray capacitance in the machine tool limited its access to the minimal discharge energy with a short duration and high frequency, even if removing the capacitors in the main power loop. That is, the ineliminable stray capacitance plays a key role in reducing discharge energy. Ashwin and Muthuramalingam [9] claimed that a transistor-type pulse generator can generate high-frequency discharge energy. Chu et al. [11] pointed out that the device response time spent in intricate detection, power amplification, and complex control circuits in transistor-type pulse generator prevents the pulse energy from further reducing. In principle, the existing pulse power generation methods all have unavoidable problems in reducing the discharge energy. As a result, new discharge energy generation theory has to be studied to reduce the discharge energy per pulse to meet the needs of micro-EDM. On the other hand, high-speed rotation of the tool electrode is commonly required in micro-hole machining to improve debris removal and assure the hole’s roundness. The brush must be used to connect the stationary pulse power supply to the rotatory tool electrode [12]. The delicate brush might be abraded or destroyed due to the direct contact of pulse-type electricity to the conductive main spindle. The greater the frequency and rotation speed, the larger the brush loss. This brings hidden dangers to the use of high-frequency pulse in micro EDM. As a result, researchers began to focus on new pulse power supply generation theory and the brushless energy transfer method in micro EDM.

In terms of new discharge energy generation methods, Abbas et al. [13] and Koyano et al. [14] presented a structure with a feed capacitor linked in series with a circuit consisting of a pulse power supply, tool electrode, and workpiece. When the pulse power supply is turned on, induced charges can be induced on the feed capacitor and the tool electrode and the workpiece simultaneously. Breakdown and discharge can occur if the tool electrode and the workpiece are sufficiently near to each other at this time. Then, the pulse power supply’s two ends are immediately connected, allowing the feed capacitor to serve as the pulse power supply, resulting in the reverse breakdown discharge between the workpiece and tool electrode [13]. Furthermore, Yahagi et al. [15] employed a 50,000 rpm spindle electrode as one plate of the feed capacitor, enabling the induced brushless power supply. However, according to Zou et al. [16], the polarity effect dominates the micro-EDM process. Although bipolar power supply machining improves deionization and discharge stability [13], it is still detrimental to machining efficiency and electrode wear. Furthermore, the feeding capacitor and the inter-electrode discharge gap must be precisely regulated, as must the matching of the pulse on and short-circuit sequencing control across the power supply.

The electrostatic field-induced electrolyte jet EDM was proposed by Zhang et al. [17]. This approach removes material by plasma breakdown of a high-voltage DC electrostatic field between the flexible jet electrode and the workpiece, and utilizes the induced charges on the jet tip and workpiece as discharge energy. Zhang et al. [18] also confirmed that by modifying the electrolyte concentration and electric parameters, the single pulse energy can be lowered to less than 10 × 10^−6^ J. However, this pulse energy is difficult to transfer to conventional EDM since it is tightly coupled with the jet electrode generation process.

This work proposes a structure and tries to decouple this energy from the jet electrode to apply it into conventional micro EDM based on previous research [17,18]. The paper is organized as follows: Section 2 constructs a decouple test platform; Section 3 explains this decouple theory and the working principle of the machining process as well as the equivalent circuit. The experimental results reveal the influence law of the E-Jet electrode discharge on the discharge of the solid electrode side. The single-point and groove machining experiments in conventional EDM with the proposed method are completed to validate its machining ability. Finally, the contents are summarized and conclusions are drawn in the final section.

## 2. Construction of Test Platform

An experimental platform is established to apply the E-Jet discharge energy generation method into conventional micro EDM, as shown in Figure 1. This platform is primarily made up of four components: the E-Jet EDM component, the traditional EDM component, the current and voltage sensing component, and the workpiece. This workpiece is made up of two parts: the sacrificial one and the right one to be machined, which are joined together by a conductive block. The first E-Jet EDM section is on the left side of the workpiece and consists primarily of a syringe with a flat pipe-like spray nozzle filled with the electrolyte, as well as a 3D manual motion platform (A) used to adjust the position of the nozzle relative to the sacrificial work-piece. One terminal of the high-voltage DC power supply is connected to the syringe’s metallic needle. The conventional EDM section is on the right side and is nearly a mirror image of the first component. The only difference lies in that the syringe and flat-headed nozzle do not contain any electrolytes. Instead, a solid tungsten needle with a diameter of 0.3 mm is placed inside the nozzle, extending out as the solid electrode. By adjusting the 3D motion platform (B), the gap between the solid electrode and the workpiece can be regulated. On this side, the other end of the high-voltage DC power supply is connected to the solid tungsten needle. In the experiment, Spellman SL60 (Spellman High Voltage Electronics Corporation, Hauppauge, NY, USA) module is selected as the high-voltage DC power supply. The third sensing component, consists of two differential voltage probes and a current sensor. The current sensor (Tektronix current sensor TCP0030A, TEKTRONIX INC., Beaverton, OR, USA) measures the discharge current, and the voltage between the electrostatic field-induced jet and the sacrificial electrode, as well as the voltage between the solid electrode and the workpiece, is measured by two Tektronix high voltage differential probes (P6015A, TEKTRONIX INC., Beaverton, OR, USA). During the discharge process, the Tektronix oscilloscope (MDO3104, TEKTRONIX INC., Beaverton, OR, USA) is used to collect current and voltage signals online.

The schematic representation of the experimental apparatus is shown in Figure 2. The power supply is a high-voltage DC with a voltage range of 0–10 kV. It can generate a strong electric field, which drags the electrolyte jet from the flat-headed nozzle toward the sacrificial electrode without any pump [18]. With the electrolyte jet extending, the field intensity increases and results in the plasma breakdown. Due to the series connection, the discharges between the solid electrode and the workpiece and between the E-Jet and workpiece almost occur in the meantime. Materials are removed on both sides of the workpiece. Following the discharge, the surface tension can pull back the electrolyte jet towards the nozzle, leading to the discharge interruption and forming a pulse interval. As a result, periodic spontaneous discharges can occur.

On the E-Jet EDM side, the discharge energy is generated by the induced charges on the jet surface and the workpiece. The dynamic imbalance of the electric force of the induced charges and electrolyte surface tension produces the electrostatic-induced pulsed jet, which is used as the tool electrode. As a result, the generation of jet electrodes and that of discharge energy are always tightly coupled. This E-Jet EDM varies greatly from electrolyte jet machining (EJM), which uses a high-pressure pump to force electrolytes onto components to remove material by electrochemical machining, as described by Liu et al. [19]. Furthermore, it is also different from the electrochemical discharge machining (ECDM) proposed by Goud et al. [20] and Singh and Dvivedi [21], in which bubbles generated by electrochemical reactions gather around the tool electrode, and the material is etched by the discharge generated by the breakdown of the gas bubble.

In Figure 2, the pulsating discharge of the solid electrode and the workpiece in conventional EDM can be controlled by the dynamical E-Jet EDM by joining the sacrificial electrode to the workpiece. The pulsating discharge of the E-Jet EDM can be used to control the traditional EDM process. The power source is a high-voltage DC voltage power supply without any external other pulse power supply or pulse control circuit, and the spontaneous periodic E-Jet discharge process can realize the periodic discharge on traditional EDM side.

## 3. Experimental Results and Discussion

### 3.1. Equivalent Circuit Analysis of the Discharge Process

The electrolyte in the nozzle is connected to the positive terminal of the high-voltage DC power supply in Figure 2, and the solid electrode is connected to the negative terminal. The sacrificial workpiece and the connected workpiece between the nozzle and the solid electrode are suspended in potential.

At first, the jet dragged from the nozzle is induced with positive charges on its surface by the high-voltage DC electrostatic field, while the solid electrode induces with negative charges. As shown in Figure 3a, the negative charges are induced on the surface of the sacrificial workpiece opposite the jet end, while positive charges are induced on the surface of the workpiece opposite the solid electrode. The total amount of generated positive and negative charges on these two sides of workpiece should be identical, but the distributions are different in space and polarity. The micro-E-Jet is emitted from the cone tip when the electric field force of induced charges on the Taylor cone at the nozzle outlet exceeds the liquid surface tension, which is similar to the phenomenon observed by Rosell-Llompart et al. [22], causing the gap between the jet tip and the workpiece to narrow. The E-Jet discharges with the sacrificial workpiece because the electric field strength between the E-Jet tip and the workpiece is larger than the air breakdown voltage [17], as illustrated in Figure 3b. The jet and plasma appear to be connecting the nozzle to the workpiece at this moment. The power supply voltage is then delivered between the solid electrode and the workpiece, causing the charge distribution on the workpiece surface to rapidly gather. As shown in Figure 3c, the inter-electrode voltage rises sharply, causing discharge breakdown between the solid electrode and the workpiece, resulting in traditional EDM in gas. At that point, both the discharges between the E-Jet and the workpiece and between the solid electrode and the workpiece are active. If the plasma impedance is ignored, it can be treated as if two ends of the power supply are connected. The discharge reduces the induced charges on the jet surface quickly. The jet retreats to the nozzle outlet as the electric field forces on it decrease and become less than the surface tensions. As shown in Figure 3d, the discharge between the solid electrode and the workpiece begins to extinguish at this point. Soon after, the cone, solid electrode, and surfaces on both sides of the workpiece resume charging in the strong electric field. As a result, the electrostatic field-induced jet, solid electrode, and workpiece surface discharge spontaneously to form a periodic continuous discharge process. As a consequence, the usage of the E-Jet EDM energy generating technology in conventional EDM can be realized indirectly. Because the field jet’s discharge energy may potentially be regulated to a very tiny scale [18], it can circumvent the discharge energy constraint in conventional micro EDM with the relaxation pulse generator [10] and the transistor-pulse generator [11]. This technology may provide an innovative method of pulsating discharge energy to micro EDM. Furthermore, this non-contact indirect electrostatic induction method provides a new solution to brushless energy transmission to high-speed spindles in micro-EDM.

### 3.2. Factors Affecting Discharge Energy

Before using the field-induced jet discharge energy, the voltages and currents on the E-Jet electrode side and solid electrode side should be considered. Moreover, to disclose the induced discharge mechanism of the solid electrode side and verify the equivalent circuit built in the last section, the sequency of these two discharges should be taken into consideration. Furthermore, to control the EDM process, the distance between the solid electrode and the workpiece, as well as the distance between the E-Jet electrode and the workpiece, should be studied, which can build a foundation for gap servo control during the EDM process.

#### 3.2.1. Discharge Waveforms Analysis

The key to distinguishing these serials’ structure discharges process is to study the waveforms of currents and voltages. To observe these discharges, a higher open circuit voltage is used to make them more visible and severe. During the discharge process, the voltage between the E-Jet nozzle and the sacrificial electrode has experienced a sharp breakdown drop to the lowest point, a stable maintaining discharge process, a slow charging rise, and a rapid exponential charging rise to the open circuit voltage, as shown in Figure 4. However, the voltage between the solid electrode and the workpiece rises from zero to 400 V and then gradually falls back to zero. The inter-electrode current rises sharply during the discharge period, then decreases and stays in a plateau for a short period before returning to zero after the discharge.

When compared to the experimental results of Ashwin and Muthuramalingam [9] and Kunieda et al. [23], this discharge current waveform appears to be a conventional pulse discharge waveform. Because the breakdown involves a time lag in which the initial electrons are accelerated to form the electron avalanche, Kunieda et al. [23] noticed that the discharge happens after the ignition delay period. When the induced charges are consumed, the current quickly drops to zero, causing a decrease in electric field forces and the jet to roll back, resulting in the discharge being extinguished. However, the discharge voltage differs from that measured in classical EDM [24]. During the E-Jet discharge process, a very sluggish voltage rising process has emerged. This is mostly due to the back-off period following the discharge of the E-Jet tip. In this process, the surface tension forces outweigh the electric field forces, leading to retreating of the jet. Some induced undischarged charges remain on the jet surface during this period; nonetheless, the charges on the E-Jet surface do not increase throughout this whole retreating phase. Simultaneously, the opposite remaining induced charges of the same magnitude are still on both sides of the workpiece, gradually approaching balance. As a result, the voltage between the solid electrode and the workpiece decreases gradually to zero in the meantime. Moreover, the time it takes for the voltage to slowly rise between the E-Jet and the workpiece is the same as the time it spends for the voltage to slowly fall between the solid electrode and the workpiece.

#### 3.2.2. Discharge Sequence Analysis

Two types and eight groups of experiments were designed to capture the waveform of a single discharge and count the sequence of discharges in order to understand the relationship between the discharges on both sides. the gap between the E-Jet nozzle and the sacrifice workpiece surface was fixed at first, and the distance between the solid electrode and the workpiece randomly adjusted. The discharge signals were sampled at four-level distances. In the second group, the distance between the solid electrode and the workpiece was fixed, and the distance between the nozzle and the sacrifice workpiece randomly regulated. All discharge signals are collected and a histogram was created, as shown in Figure 5. The abscissa represents the leading time difference of the E-Jet electrode EDM and the solid electrode EDM.

The discharge at both ends occurred almost simultaneously during the experiment. However, according to the sampling data analyzed by Matlab software, as shown in Figure 5, the E-Jet electrode EDM is mostly ahead of 1–2 discharge sample units before the solid electrode EDM, that is, about (4–8) × 10^−6^ s. This occurrence sequence is independent of the control mode, regardless of whether the solid electrode or the E-Jet nozzle is adjusted. This reveals that the E-Jet EDM process induces the discharge of solid electrode. Moreover, if retaining the distance between the E-Jet and the workpiece, adjusting the position of the solid electrode relative to the workpiece can result in discharge. Furthermore, if fixing the distance between the solid electrode and the workpiece, adjusting the gap between the E-Jet nozzle and the workpiece can also result in discharge. As a result, this method supports two types of gap servo control.

### 3.3. Influence of Distance on Discharge Energy

The gap servo control is the primary selected method in the traditional EDM to ensure stable continuous processing. As a result, the influence of the distance on the discharge energy should be considered. Because of the high-frequency noises in the high-voltage discharge process, it is difficult to calculate the discharge energy accurately and directly. We use a windowed filter to deal with high-frequency signals before calculating the discharge energy. Because the discharge energy is roughly equal to the heat generated during the discharge process [25], it can be expressed as follows:(1)$$E_{e}=\int_{0}^{t_{e}}u_{e}\left(t\right)\cdot i_{e}\left(t\right)\cdot dt$$
where *U*_e_ denotes the discharge voltage, *I_e_* denotes the discharge current, and *t_e_* denotes the discharge duration. If described more precisely, this formula can be rewritten as:(2)$$E_{e}=\overset{t_{e}/\Delta t}{\underset{0}{\sum }}u_{e}\left(t\right)\cdot i_{e}\left(t\right)\cdot \Delta t$$
where Δ*t* is the sampling interval.

#### 3.3.1. Influence of the Gap on Solid Electrode Side on Discharge energy

At first, the distance between the E-Jet nozzle and the sacrificial workpiece was set to 0.5 mm and the distance between the solid electrode and the workpiece surface to 0.195 mm. The concentration of the electrolyte (NaCl) was 5% by weight. Then, 2.8 kV was applied between the E-Jet electrode and the solid electrode. All other machining parameters were maintained as constant, and the distance between the solid electrode and workpiece was changed progressively by manually adjusting the 3D platform (B) knob in Figure 2 from 0.195 mm to 0.18 mm at 0.005 mm intervals and prohibiting the discharge from arcing or breaking off. The EDM discharge energy on the solid tool electrode side was calculated by recording the voltage and current at various discharge sites. The sacrificial workpiece and the workpiece in the experiment were both silicon, allowing the machining features to be visible to the naked eye.

When the distance between the jet nozzle and the workpiece stayed constant, as illustrated in Figure 6a, the discharge duration time decreased with increasing distance between the solid electrode and the workpiece. The discharge current dropped as the distance between the electrodes increased, as illustrated in Figure 6b, ranging from 0.1 A at 0.18 mm to 0.04 A at 0.195 mm. The gap voltage between the solid electrode and the workpiece steadily grew at a low level throughout the discharge process while the gap was less than 0.185 mm, but it quickly increased from 0.185 to 0.19 mm, from dozens of volts to 230 V, as shown in Figure 6c.

The discharge energy may be computed indirectly using Equation (2). The discharge energy between the solid electrode and the workpiece, as well as the discharge energy between the E-Jet and the workpiece, decreased steadily as the gap distance between the solid electrode and the workpiece increased, as shown in Figure 7. On the jet side, the discharge energy was always larger than that on the solid electrode side. The narrower the gap distance between the solid electrode and the workpiece, the bigger the discharge energy difference between the jet electrode side and the solid side.

These might be caused by the reasons that in the discharge process, the capacity between the solid electrode and the workpiece can be treated as the feed capacity to that between the jet end and the sacrificial electrode, which in turn impacts the discharge between the solid electrode and workpiece. Changing the distance between the solid electrode and the workpiece entails changing the voltages between the jet end and the workpiece, as well as between the workpiece and the solid electrode. When the distance between the solid electrode and the workpiece is reduced, the capacitance between the solid electrode and the workpiece rises. Although the capacitance between the jet nozzle and the workpiece remains constant, the total capacitance between the jet nozzle and the solid electrode increases. At this stage, the strength of the electric field rises. The charge density on the cone surface increases as the electric field intensity increases, resulting in an increase in electrostatic repulsion forces at the cone tip, causing the cone angle to expand. In addition, the cone at the nozzle outlet begins to recede, resulting in a shorter cone. Meanwhile, the increased repulsive force causes the jet to expand in diameter. Heikkilä et al. [26] claimed that reducing the tip-to-collector distance can alter the form of the cone/jet in the nanofiber spinning process by the changes of the electric field. The electrolyte jet, on the other hand, can be regarded as a conductor in the E-Jet EDM process, allowing for discharge rather than continuous spinning. The higher the intensity charge density, the longer the discharge duration (Figure 6a) and the higher the current (Figure 6b). As the distance grows, the surface charge density falls gradually, the electrostatic repulsion on the cone’s surface lowers gradually, the cone’s length increases gradually, the cone angle decreases gradually, and the initial diameter of the jet decreases gradually. A drop in surface charge density and a small diameter of the jet result in a very short duration, and a minor charge depletion leads to a decrease in electric field forces and results in jet retraction. Because the interelectrode voltage is directly connected to the plasma characteristics during the discharge process, the greater the maintenance voltage, the larger plasma discharge gap [27]. Simultaneously, lowering the charge density and jet diameter decreases the strength of the discharge between the jet and the workpiece, causing the voltage to progressively rise as the electrode moves away from the workpiece, as shown in Figure 6c. Furthermore, when the discharge distance of the solid electrode jet electrode increases, the surface induced charge density of the two electrodes progressively decreases, which explains why the discharge energy reduces with increasing distance.

As shown above, when the distance between the jet electrode and workpiece is fixed, the gap servo control of classical EDM may be modified in real-time to create steady EDM. At the moment, the pulsating jet electrode serves as a pulse power switch for the solid electrode discharge.

#### 3.3.2. Influence of E-Jet EDM Gap on Discharge Energy

In the experiment, fixing the distance between the solid electrode and the workpiece, the influence of E-Jet discharge on the discharge energy of solid electrode EDM was explored by adjusting the distance between the jet nozzle and the workpiece. The electrolyte content was 5% by weight. The voltage was 2.8 kV. Gradually, the distance between the jet nozzle and the workpiece was reduced from 0.69 mm to the workpiece surface at 0.02 mm intervals. The knob of the motion platform (A) was adjusted to achieve 0.69 mm, 0.67 mm, 0.65 mm, and 0.63 mm, as illustrated in Figure 1. the changes in voltages and currents at various discharge sites were recorded, ensuring that both ends can discharge. It was explored how the distance between the jet nozzle and the workpiece influences the current, voltage, discharge duration, and energy distribution between the solid electrode and the workpiece.

The graph in Figure 8a demonstrates that the duration of the jet discharge gradually reduced as the distance between the jet end and the workpiece surface rose, meaning that the discharge pulse width decreased as the distance increased. At the same time, as seen in Figure 8b, the discharge current steadily diminished. The voltages between the E-Jet nozzle and the workpiece gradually rose as the distance between the E-Jet nozzle and the workpiece grew, as illustrated in Figure 8c. The voltage on the solid side is much lower than the voltage on the E-Jet side.

The discharge energy on these two sides may be estimated using Equation (1). It is revealed that the discharge energy on the E-Jet side reduces as the distance between the nozzle and the workpiece increases. Meanwhile, as seen in Figure 9, the discharge energy on the solid side declined extremely slowly.

In this process, the jet end and the workpiece were utilized as a feed capacitor when the distance between the solid electrode and the workpiece was maintained constant and the distance between the jet nozzle and the workpiece was adjusted to discharge. The jet nozzle and the workpiece can be treated as a variable capacitor at this point, whereas the solid electrode end and the workpiece were a fixed capacitor.

The field intensity in between steadily increased as the distance between the jet nozzle and the workpiece decreased. At this point, the jet’s surface charge density rose, causing the jet cone angle to rise, the cone length to fall, and the initial jet diameter to rise. Increasing charges and jet width caused a strong discharge, which increased discharge duration (Figure 8a) and discharge current (Figure 8b). This procedure is similar to that when changing the solid electrode to initiate the discharge shown in Figure 6a,b. As the distance grew, the surface charge density fell, resulting in a decrease in discharge energy, as shown in Figure 9. As the distance decreased, the larger E-jet diameter and the intensive charge density resulted in the violent plasma, gradually leading to the decrease of the maintaining voltage between the solid electrode and workpiece and the E-jet tip and the workpiece, as shown in Figure 8c.

It can be seen from these two control approaches that the discharge between the solid electrode and the workpiece may be controlled independently of the relative distance between the jet end and the solid electrode or the distance between the solid end and the workpiece.

### 3.4. Machining Experiments

Experiments using single-point and groove machining were carried out to confirm the processing capacity of this technology. The solid electrode was a tiny tungsten electrode machined by electro-chemical etching method proposed by Kang and Tang [28], and the E-Jet electrolyte was a 5 wt.% NaCl aqueous solution. Polished silicon (Guangzhou Fangdao Semiconductor Co., Ltd., Guangzhou, China) was chosen as the workpiece due to its semiconducting properties, and the mirror surface allowed the machined crater and groove to be roughly observed by the naked eye before using a high magnification microscope, which is suitable for initial machining capability verification experiments.

The distance between the nozzle that generates the E-Jet and the sacrificial workpiece during the discharge process was 0.62 mm, and the distance between the solid electrode and the workpiece was 0.18 mm. The applied voltage was 2.8 kV between the nozzle and the solid electrode. Table 1 lists the machining parameters. After 1 min of processing, a point feature on the workpiece surface could be found by the naked eye. The cave could be observed through a microscope (VHX-6000 microscope, KEYENCE CORPORATION, Osaka, Japan) after being ultrasonically cleaned for 20 min and dried, as shown in Figure 10. It was discovered that a pit with a diameter of 20 μm and a depth of 9 μm could be machined. On the bottom of the pit, it was possible to see a discharge erosion phenomenon. This demonstrates that the energy generation method can be used in the solid electrode discharge process and that the materials can be removed. At the same time, the pit’s shape differed a little from that of the cylindrical solid electrode with a tip in conventional EDM. This is primarily due to the fact that the solid electrode did not rotate, and the gap distance reached 0.18 mm, which is significantly greater than that in traditional EDM in gas observed by Wang and Shen [29]. Furthermore, due to the large gap, plasma drift occurred during discharges, resulting in the irregular shape of the machined pit [30].

The machining parameters, solid electrodes, and workpieces used in the groove machining experiment were the same as that in previous pit etching process. The discharge gap between the jet electrode and the sacrificial workpiece remained constant, as did the gap between the solid electrode and the workpiece. The only difference was that the manual platform (B) under the solid electrode was slowly moved in the horizontal plane, and perpendicular to the electrode direction, to test the method’s continuous groove processing feasibility. Figure 11 depicts the finished groove observed by VHX-6000 microscope. It was discovered that a regular groove with a depth of 4μm and a width of 30 μm could be processed. Although the width of the groove was fairly uniform, it was still significantly larger than the diameter of the solid electrode. This is primarily due to the plasma deflecting in a wider gap in the gas.

The experiments above show that the E-Jet EDM-based periodic discharge energy generation strategy can be decoupled and applied into the traditional EDM process. Periodic discharge can be generated automatically without the use of a traditional RC-type or transistor-type power supply. Furthermore, if the solid electrode rotates, the uncontacted E-Jet induced discharge used in traditional EDM can provide a solution for pulse power supply in micro EDM without the use of brushes.

## 4. Conclusions

By incorporating the E-Jet EDM energy generation method into the solid electrode EDM process, a new non-contact discharge energy generation method for traditional micro EDM was developed. This method has a wide range of potential applications in the EDM process. The study yielded the following useful findings:1)The electrostatic induction discharge energy can be decoupled from the E-Jet EDM process to realize its application into the traditional EDM by connecting the electrical field-induced electrolyte jet EDM and the traditional EDM in series in structure.2)It was discovered that the discharge on the E-Jet EDM side is 1–2 sampling units ahead of the discharge on the solid electrode EDM side, proving that the discharge process is induced by the E-Jet process.3)The discharge signals analyzed demonstrated that the solid electrode EDM can be controlled by adjusting the gap between the solid electrode and the workpiece with the pulse discharge energy controlled by E-Jet EDM method. If fixing the distance between the E-Jet and the workpiece, the discharge energy per pulse will decrease with the increasing distance of the solid electrode from the workpiece.4)The discharge energy of the solid electrode EDM can also be controlled by adjusting the gap between the E-Jet electrode and the workpiece surface, allowing the energy to be indirectly used in the solid EDM process. The discharge energy decreases with the increasing the distance between the E-Jet and the workpiece.5)Experimental results show that this method can process a pit with a diameter of 20 μm and a depth of 9 μm and groove with a depth of 4 μm and a width of 30 μm on silicon wafers, and the results demonstrate the method’s effectiveness.

The method of decoupling pulse energy from the E-Jet EDM process using two discharge devices connecting in serials proposed in this study can be applied to the energy supplying of traditional micro EDM. It can provide discharge energy by means of electric field induction without using brushes. In this way, the rotation speed of the micro electrode can be set higher, and there is no friction or contact resistance in the power supply circuit. The experimental platform will be established, and related studies will be carried out in the future.

## Figures and Tables

**Figure 1 materials-16-03963-f001:**
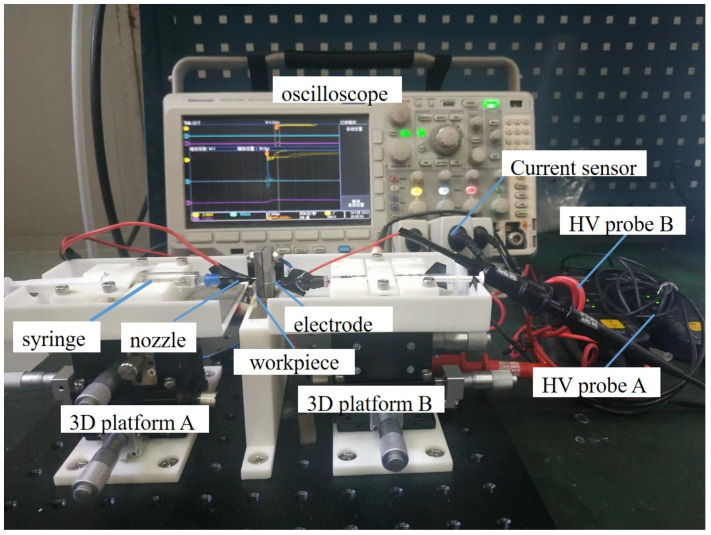
The experimental platform of the E-Jet discharge energy used in traditional EDM.

**Figure 2 materials-16-03963-f002:**
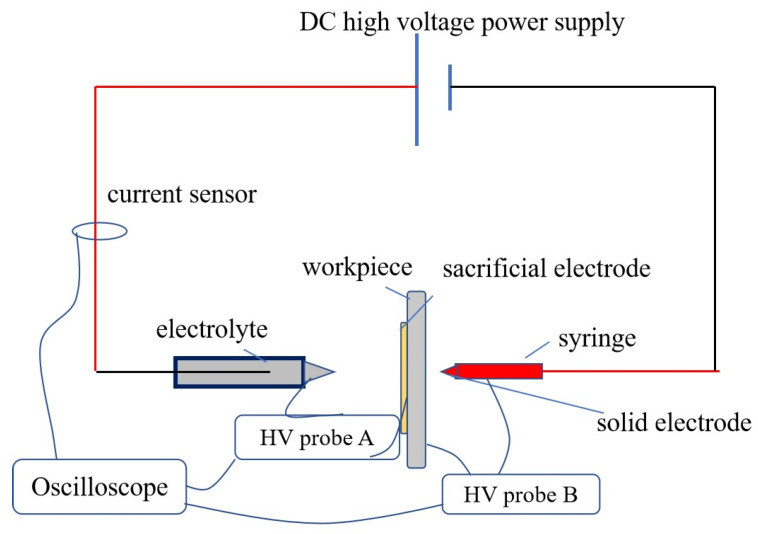
Schematic diagram of energy generation principle of micro EDM based on E-jet process.

**Figure 3 materials-16-03963-f003:**
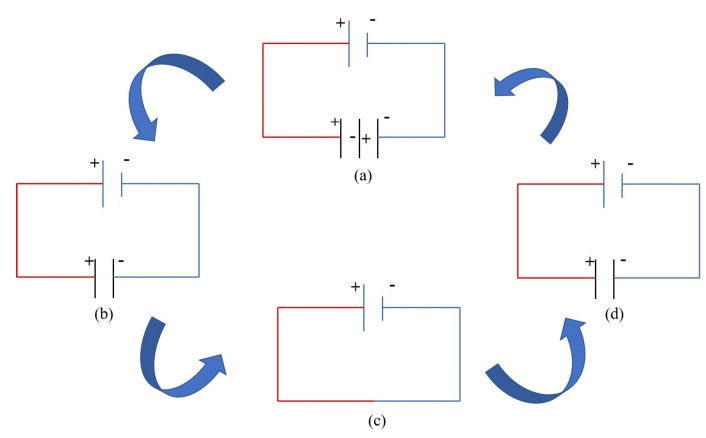
Decomposition schematic diagram of E-Jet energy generation method in micro EDM: (**a**) the initial electric field distribution state; (**b**) the equivalent electric field distribution after an E-Jet discharge generation; (**c**) the equivalent electric field distribution when both sides of dis-charges are active; (**d**) the equivalent electric field distribution when the E-jet retreats with only solid electrode discharge remaining.

**Figure 4 materials-16-03963-f004:**
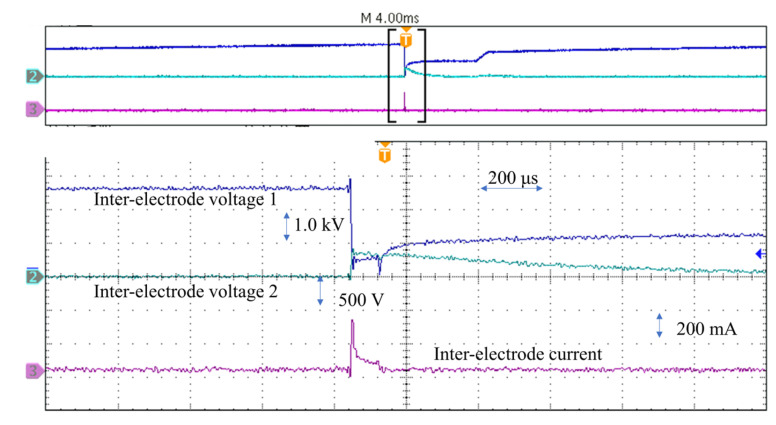
The voltages and currents waveforms during the discharge processes.

**Figure 5 materials-16-03963-f005:**
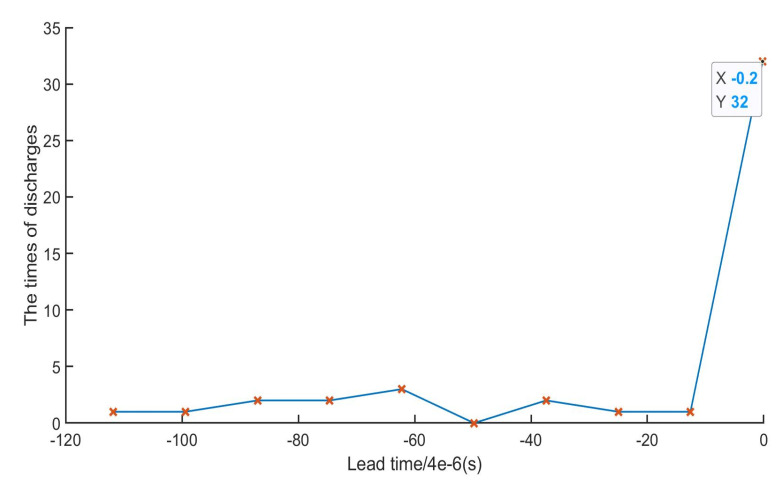
Time difference of discharge process between E-Jet EDM and solid electrode EDM.

**Figure 6 materials-16-03963-f006:**
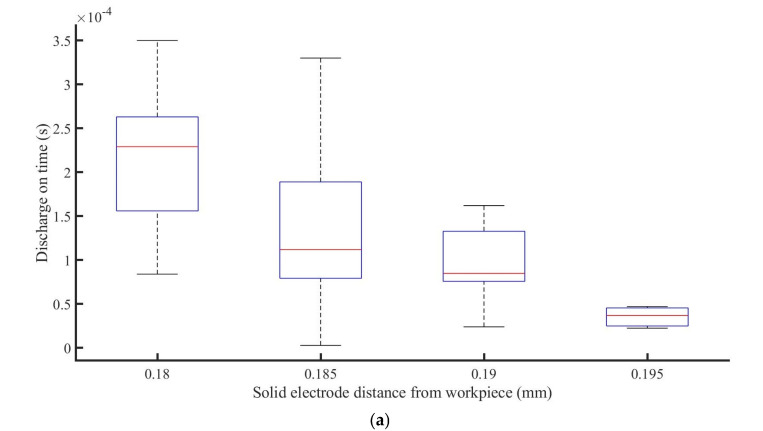
Analysis of the discharge duration (**a**), gap current (**b**), and gap voltage (**c**) changes with the variation of the distance between solid electrodes and workpiece.

**Figure 7 materials-16-03963-f007:**
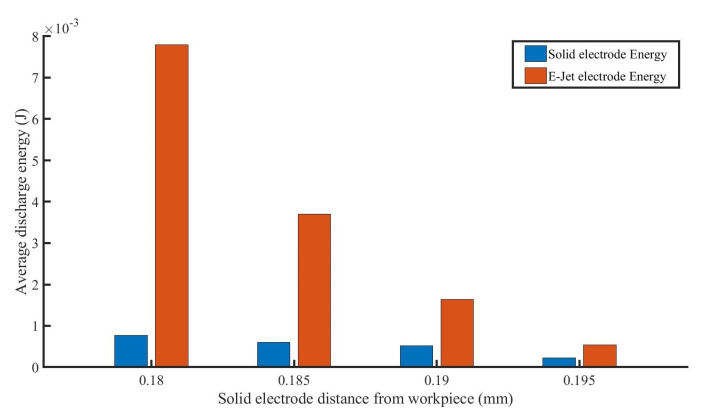
The discharge energy changes with the variation of the distance between solid electrodes and workpiece.

**Figure 8 materials-16-03963-f008:**
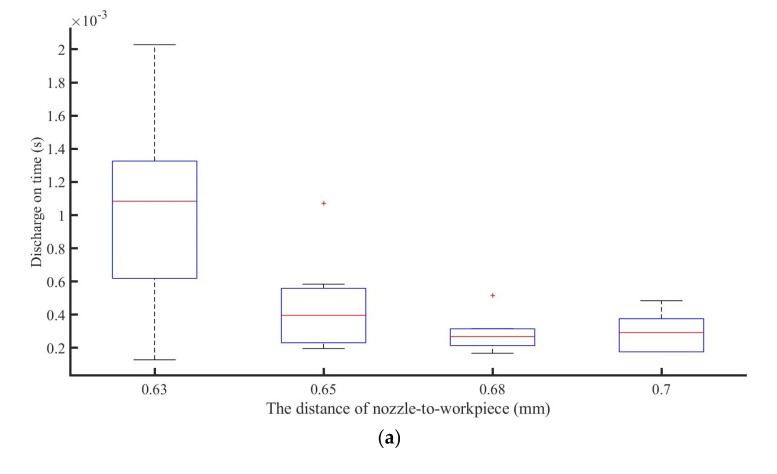
Analysis of the discharge duration (**a**), gap current (**b**), and gap voltage (**c**) change with the variation of the distance between E-Jet nozzle and workpiece.

**Figure 9 materials-16-03963-f009:**
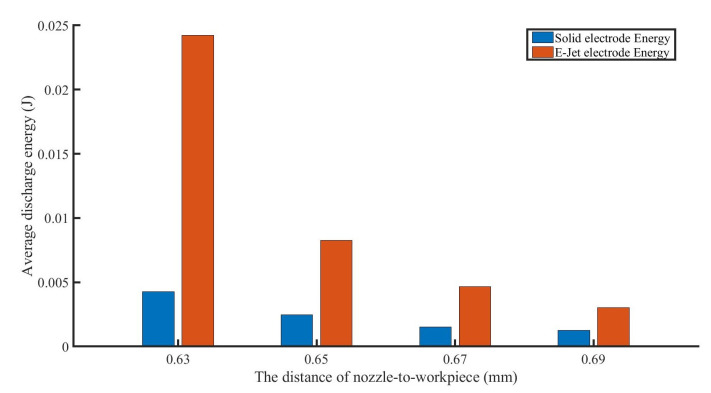
The discharge energy change with the variation of the distance between E-Jet nozzle and workpiece.

**Figure 10 materials-16-03963-f010:**
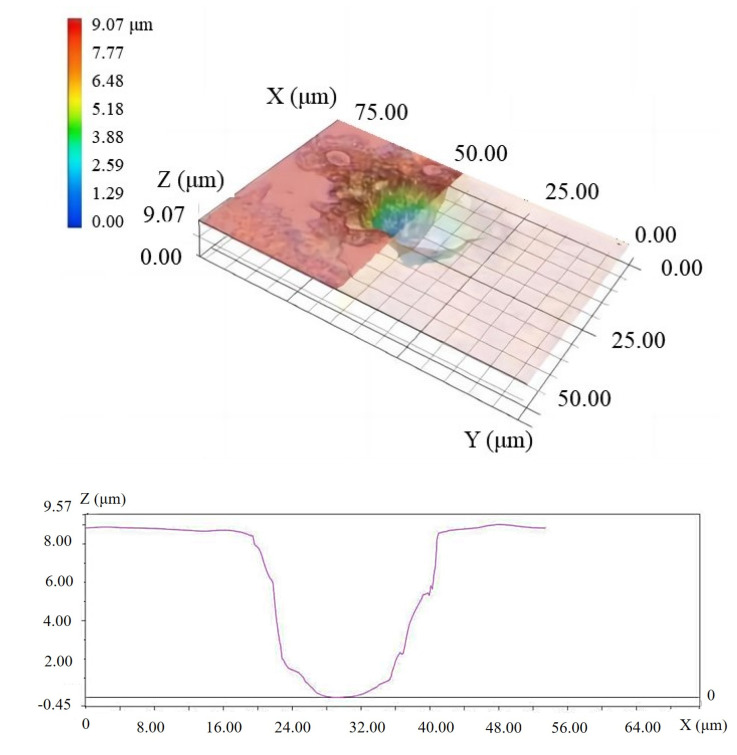
Analysis of the pit morphology machined by EDM in gas using solid electrode.

**Figure 11 materials-16-03963-f011:**
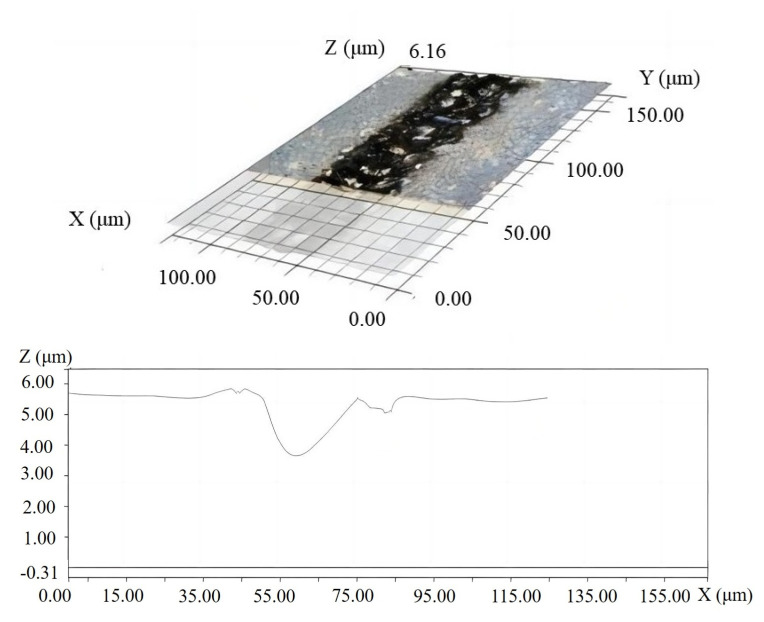
Continuous groove produced by EDM in gas using solid electrode.

**Table 1 materials-16-03963-t001:** Machining parameters.

Parameters	Value
Solid electrode	Tungsten
Electrolyte jet electrode	5 wt.% NaCl
Workpiece	Polished silicon
Voltage	2.8 kV
E-Jet gap distance	0.62 mm
EDM gap distance	0.18 mm

## Data Availability

Not applicable.

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
