# Peer review of "Research on Electrostatic Field-Induced Discharge Energy in Conventional Micro EDM"

_materials, 2023, doi:10.3390/ma16113963_

Round 1
Reviewer 1 Report
The article presents an interesting and current topic. At present, efforts are still being made to improve the efficiency of the EDM process, in terms of machining speed, but also to reduce the wear of the tool electrode. The analyzed research material was correctly presented. However, the article requires a few changes to be introduced in order to increase its value and the interest of Readers. Suggestions for consideration are given below:
1. The Introdution part could be more extensive, and include more information about the EDM process (its advantages, limitations, what is currently being worked on to optimize this process).
2. I suggest to include the part "Machining experiments" in the chapter "Results and discussion". Here, I also propose to include a chapter such as "Results and discussion" in the article. This chapter is one of the key chapters of the article.
3. In "Machining experiments", please provide in Table 1, all process parameters (including current amplitude, machining gap thickness, etc.), the result parameters could also be included in this Table.
4. Some graphs are too large (Figure 7) and some are hard to read (Figure 10). I suggest to refine the quality of Figures.
5. In the Conclusions section, please also refer to further research in the analyzed topic.

Author Response
Dear reviewer,
The authors would like to thank you for your prompt processing of our manuscript. It is a great honor for us to receive your preliminary approval for our work. We take this opportunity to revise our manuscript very seriously. We also deeply appreciate the reviewer’s professional and meaningful comments on our previous draft. It is in the process of thinking and answering these questions that we have enriched our manuscript and made it more rigorous.
Specifically, we have carefully discussed and studied the comments raised by the reviewers one by one. The explanations for these questions are listed in detail in the responses below. Based on the reflections on these issues, we have prepared a revised manuscript. More clarifications are added and some expressions are rephrased. All these changed parts are marked in blue in our revised manuscript.
Each response listed below consists of two parts, a "Response" section and a "Revise" section. The "Response" section contains an explanation to the question raised by the reviewer, while the "Revise" section lists specific changes of the manuscript in italics, including expressions before and after revision.
The attachment below is our responses to the reviewer’s comments.
Thanks for all the help.
Best regards.

Reviewer 2 Report
1. What is the main question addressed by the research? 2. A detailed methodology in order to repeat the experiments is advisable. 3. Conclusions should be made more stronger.4. Please add micron markers in Figs. 10 and 11.
5. Please define the x- and y-axes in Figs. 10 and 11 (last part).
6. The references are numbered twice. Please check.
7. Please add some ballpark numbers in conclusions to make it quantitative.
8. What does Fig. 5 represent? Time versus lead time? Please check. Please explain what is it?
9. Explain the novelty of the current work.
10. Improve the quality of Figure 6 a,b, c and 8 ab c ,10 (last part) and 11 (last part)
11. There are many typos in the manuscript. For example, in Page 7 line 262. If describing the formula more practically, it can rewrite as… to be changed as … If describing the formula more practically, it can be rewritten as…
12. The authors can improve the discussion of the paper. They can refer to and cite the following works.
https://doi.org/10.1533/9780857095145.2.202, https://doi.org/10.1177/14644207211058297, https://doi.org/10.1088/2051-672X/ac8757.
13. Reference from a single country/ region is not advisable. Equal chance to be given for authors from all parts of the globe. Authors must try to avoid self-citations as well. This will be viewed seriously upon revision.
Minor English corrections are required.
Author Response

(The authors gave the same response as above.)

Round 2
Reviewer 2 Report
Accept